# Computational Chemistry Strategies to Investigate the Antioxidant Activity of Flavonoids—An Overview

**DOI:** 10.3390/molecules29112627

**Published:** 2024-06-03

**Authors:** Yue Wang, Chujie Li, Zhengwen Li, Mohamed Moalin, Gertjan J. M. den Hartog, Ming Zhang

**Affiliations:** 1Department of Pharmacology and Personalized Medicine, School of Nutrition and Translational Research in Metabolism (NUTRIM), Maastricht University, 6200 MD Maastricht, The Netherlands; yue.wang@maastrichtuniversity.nl (Y.W.); c.li@maastrichtuniversity.nl (C.L.); gj.denhartog@maastrichtuniversity.nl (G.J.M.d.H.); 2School of Pharmacy, Chengdu University, 2025 Chengluo Avenue, Chengdu 610106, China; lizhengwen2012@outlook.com; 3Research Centre Material Sciences, Zuyd University of Applied Science, 6400 AN Heerlen, The Netherlands; maxamed.moalin@gmail.com; 4Hainan University-HSF/LWL Collaborative Innovation Laboratory, College of Food Sciences & Engineering, Hainan University, 58 People Road, Haikou 570228, China

**Keywords:** flavonoids, antioxidant activity, computational chemistry analysis

## Abstract

Despite several decades of research, the beneficial effect of flavonoids on health is still enigmatic. Here, we focus on the antioxidant effect of flavonoids, which is elementary to their biological activity. A relatively new strategy for obtaining a more accurate understanding of this effect is to leverage computational chemistry. This review systematically presents various computational chemistry indicators employed over the past five years to investigate the antioxidant activity of flavonoids. We categorize these strategies into five aspects: electronic structure analysis, thermodynamic analysis, kinetic analysis, interaction analysis, and bioavailability analysis. The principles, characteristics, and limitations of these methods are discussed, along with current trends.

## 1. Introduction

### 1.1. The Enigma of Flavonoids’ Antioxidant Activity

A major desire is to be and stay in good health. This desire has fueled our interest in compounds that have a positive impact on the body. However, the limitations and negative aspects of synthetic drugs are becoming increasingly apparent. This has revived our interest in natural compounds present in our diet that can promote health. A group of natural compounds gaining significant attention are flavonoids. A high dietary intake of flavonoids has been associated with positive effects on health, such as a lower incidence of cardiovascular diseases and cancer [1,2,3]. Based on the paradigm coined by Brown and Fraser in 1868, we assume that the fundament for this beneficial effect lies in the interaction of flavonoid molecules with molecules and processes in our bodies [4]. 

Surprisingly, after decades of research, the exact molecular mechanism responsible for the antioxidant effect of flavonoids is still enigmatic [5,6]. First of all, this is because measuring antioxidant activity appears to be less straightforward than commonly assumed. Complicating factors affecting the outcome of antioxidant assays are as follows: The myriad of reactive species used in the assays varies from ‘simple’ highly reactive oxygen radicals to complex relatively stable synthetic radicals.The degradation of the unstable compounds during the experiments.Compartmentation, e.g., of lipophilic compounds, in cell culture experiments.

Regarding the latter, it is usually tacitly assumed that in cell experiments, the concentration in all compartments is equal to the initial concentration of the compound in the culture medium, which of course is not the case. Moreover, the existence of the different types of antioxidant assays, along with the procedural differences within specific assays, blur our view on the antioxidant activity of flavonoids.

In addition, the group of flavonoids comprises more than 5000 compounds. In line with the paradigm of Brown and Fraser, we know that a relatively small change in chemical structure can lead to a drastic impact on biological activity mechanism. Nevertheless, often all flavonoids are put under the same umbrella, incorrectly assuming that the molecular mechanisms underlying the antioxidant activity of all flavonoids are nearly identical.

### 1.2. The Application of Computational Chemistry in Exploring the Antioxidant Activity of Flavonoids

An intriguing alternative for obtaining a more accurate understanding of the biological activity of flavonoids is to leverage computational chemistry. In recent years, our understanding and skills in quantum chemical theory have further deepened causing the expanding landscape of computational chemistry [7,8]. New ‘pathways’ for exploring the antioxidant activity of flavonoids continuously emerge. In this review, we will provide an overview of diverse strategies of computational chemistry used to elucidate and visualize the antioxidant mechanism of flavonoids.

## 2. Preliminary Statistics on Research Status

To begin, we gathered the literature that utilized computational chemistry to investigate the antioxidant activity of flavonoids over the past five years. In this survey, we found data on 189 flavonoids. Figure 1 illustrates the basic skeleton of the subclasses to which the studied structures belong, along with the number of compounds in each subclass that have been studied over the past five years. This underscores a predominant focus on flavones/flavonols and flavonoid-glycosides.

In this period, computational chemistry has been employed to explore five aspects of flavonoids related to antioxidant activity: electronic structure, thermodynamic enthalpies, the reaction kinetics in the fee radical scavenging reaction, intramolecular and intermolecular interaction, and bioavailability. Among these, electronic structure analysis has been the most extensively studied. Figure 2 illustrates the methods employed in the past five years to investigate the electronic structure of flavonoids, along with the proportion of each method used.

Figure 2 clearly shows that in this period, the classical methods were preferred, including the frontier molecular orbitals, molecular electrostatic potential, and global chemical reactivity descriptors. The more recently developed and intuitive indicators, for example, the density-of-states and donator–acceptor map, have not received significant attention. Apparently, most researchers prefer to follow the already well-known and well-trodden paths. However, it is crucial to acknowledge the limitations of these conventional approaches. At times, the chosen calculation methods may not align with the assumed scenario.

To progress, we have to explore and forge new paths. Ultimately, we should strive to combine diverse methods, as each provides valuable but limited information. Integrating various methods and perspectives will yield a more accurate and comprehensive understanding. 

Our goal is to optimize the use of computational chemistry in resolving the antioxidant activity of flavonoids. Therefore, the principles, characteristics, and limitations of the above five calculation strategies are discussed in this review, as well as current trends. 

## 3. Computational Chemistry Strategies to Investigate the Antioxidant Activity of Flavonoids

### 3.1. Electronic Structure Analysis

Electronic structure analysis strategies giving the electron distribution characteristics of flavonoids have often been used to explain and predict their activity, pinpoint the reaction sites, and elucidate the molecular mechanisms, as well as estimate the stability of complexes between flavonoids with targets, e.g., metals and macromolecules. In the following paragraphs, the electronic structure analysis strategies used in the past five years will be discussed successively.

#### 3.1.1. Frontier Molecular Orbitals (FMOs)

The frontier orbitals are the molecular orbitals that preferentially participate in the reactions. These include (i) the highest occupied molecular orbital (HOMO) and (ii) the lowest unoccupied molecular orbital (LUMO) [83]. The energy levels of the HOMO (E_HOMO_) and LUMO (E_LUMO_) specify the electron donating and accepting ability of a molecule, respectively. For an antioxidant, a more negative value of E_HOMO_ implies a higher potency to donate an electron to a radical [84,85]. The localization of the HOMO and LUMO also pinpoints the key pharmacophore of a molecule. The HOMO provides a reliable indication of the location of the outermost, and consequently most energetic, electron in a molecule. Theoretically, this is also the electron most readily donated by an antioxidant. In Figure 3, the HOMO orbital of the quercetin (Q) is shown, which indicates that Q is most likely to donate an electron from the B ring and the double bond in the C ring.

The energy gap of frontier molecular orbital (E_gap_) is the difference between E_LUMO_ and E_HOMO_; the higher E_gap_, the more kinetically inert the molecule is [89,90]. E_gap_ has been suggested to be a more precise descriptor than E_HOMO_ for the activity investigation of an antioxidant [91]. 

FMO analysis has been the most widely used tool to study the antioxidant activity of flavonoids over the past five years. However, FMO does not provide the full picture of chemical reactivity [92]. Recent evidence shows that the lower-lying molecular orbitals cannot be neglected [93,94,95]. Bulat et al., in 2021, used FMO to predict the reaction site of twelve molecules and only succeeded with five of them. They found that average local ionization energy I(r) is a better descriptor [83]. Average local ionization energy, introduced in 1990, is the average energy needed to remove an electron at any point of a molecule [96]. Apparently, the contribution of all electrons, rather than just the electrons in specific orbitals of the molecule, needs to be taken into account [97,98]. I(r) has been used to explore the activity of antioxidants, such as L-ascorbate and α-tocopherol [99,100]. Despite its potential advantages, I(r) has not been used to study the antioxidant activity of flavonoids in the past five years.

#### 3.1.2. Molecular Electrostatic Potential (MEP/ESP)

MEP is defined as the energy required to move one positive point charge from infinity to a specific point in a molecule. The surface MEP map depicts the electrostatic potential distribution of the molecule with different colors [101]. Typically, the red area signifies a negative MEP surface, indicating a tendency to donate electrons, while the blue region indicates a positive MEP surface, suggesting a propensity to accept electrons. The surface MEP map of Q is shown in Figure 4. 

The surface MEP map is an effective tool for predicting reaction sites and has already been applied to flavonoids [58]. There is a strong correlation between the MEP and the acidity of the molecule, indicating that the MEP can be used to estimate the pK_a_ value of the hydroxyl groups at different positions on flavonoids [102]. Given the significant influence of deprotonation on the antioxidant activity of flavonoids (as will be discussed in Section 3.5.1), this warrants further exploration.

#### 3.1.3. Global Chemical Reactivity Descriptors (GCRDs)

GCRDs describe the sensitivity of a compound’s structure to slight changes in the external potential and electrons. These descriptors are used to predict reactivity and structure stability [103]. The Hartree–Fock theory defines the energy involved in extracting and donating electrons from a molecule, as IP and EA respectively [83]. The IP is equal to the difference of electronic energy between the N state (the neutral state of the molecule) and the N-1 state (the state after losing one proton). The adiabatic ionization potential (AIP) is the electronic energy gap between the optimized N and N-1 states, while the vertical ionization potential (VIP) only uses the optimized N state. The lower the IP, the less energy is needed to lose an electron [104,105]. The GCRPs commonly used to measure the electron transfer characteristics of antioxidants are given in Table 1.

GCRPs are important, since they closely correlate with various physicochemical properties like bonding energies, aromaticity, and polarizability [113]. The advantage of its broad correlation comes with the disadvantage that the results of GCRPs usually need to be refined by more specific descriptors.

#### 3.1.4. Natural Bond Orbitals (NBOs)

The NBO theory partially localizes molecular orbitals by diagonalizing the density matrix to facilitate an intuitive understanding of electron orbitals and distribution. NBO analysis is employed in computational chemistry to calculate the electron-occupied orbitals and distribution of electron density between chemical bonds and describe the migration and rearrangement of bonds during chemical reactions [114,115]. 

Transitions and rearrangements often occur during the antioxidant processes of flavonoids, yet the precise mechanisms remain elusive due to the difficulty in capturing them. Undoubtedly, NBO analysis stands out as a powerful tools to fill this gap. Moreover, it also deserves wider utilization in exploring the impact of substituent groups. Although NBO analysis is powerful, it currently only serves as an auxiliary tool for describing electron distribution in the study of the antioxidant activity of flavonoids [47,54]. 

It should be noted that traditional NBO analysis results may be unreliable for systems with complex electronic structures and multi-center delocalization. However, an enhanced version is already available, capable of accurately describing the shifts of localized bonds in reaction processes [116]. Moreover, NBO analysis needs to be improved to examine the interaction between metals and non-metals [117]. Therefore, it is not yet suited to explore the metal-chelating activity of flavonoids. 

#### 3.1.5. Natural Transition Orbitals (NTOs)

The transition of electrons between the ground state and excited states in molecules can be described as the migration between different NTOs [79]. At present, the NTO analysis of flavonoids focuses on the outer electron going from the ground state (S_0_) to the first excited state (S_1_) [118]. In particular, the emphasis in the study revolves around excited state intramolecular proton transfer (ESIPT), the process in which excited molecules relax their energy through tautomerization, the intramolecular transfer of a proton. Chaofan et al. demonstrated, using density functional theory (DFT) and time-dependent density functional theory (TD-DFT), that the rate of the ESIPT reaction has a negative relationship with the antioxidant activity in a series of flavonoids [119]. 

#### 3.1.6. Spin Density Distribution (SDD)

SDD is a descriptor that shows the delocalization of the unpaired electron in a radical molecule. For flavonoids, it provides information on the reaction kinetics and stability of the intermediates formed during free radical scavenging. A highly delocalized SDD indicates a fast radical scavenging reaction and high stability of the radical intermediate. Here also, intramolecular proton transfer should be considered together, as it can assist the unpaired electron to delocalize and disperse over the molecule [120,121]. For example, as shown in Figure 5, for Q-7-O-radical, a long-conjugated chain can be formed after intramolecular proton transfer, the lone pair electron is delocalized throughout the molecule.

SDD holds a unique position in the study of flavonoids’ ability to directly scavenge free radicals because it actually describes the intermediates formed. It is often used in conjunction with other reactivity descriptors to comprehensively evaluate the antioxidant activity of flavonoids.

#### 3.1.7. Polarity and Dipole Moment

The uneven distribution of charges in covalent bonds of a molecule creates a dipole moment and makes the molecule polar. This uneven distribution is caused by the difference in electronegativity between bonded atoms. The extent of the imbalance is given by the MPI [122].

Although polarity and dipole moment are not descriptors that directly describe the electron distribution, they are also often calculated when studying the activity of antioxidants, as they are a ‘conventional characteristic’ of a molecule [123]. The dipole moment has also been used to study the stereo-conformation and electronic distribution properties of flavonoids [124,125].

#### 3.1.8. Fukui Function (FF)

In computational chemistry, the FF describes the changes in electron density in frontier orbitals caused by the change in the total number of electrons in a molecule [126]. 

FF descriptors have been developed separately for nucleophilic reactions (*f*^+^), electrophilic reactions (*f*^−^), and/or radical reactions (*f*^0^), as shown in the following formulas:$$f^{-}=\rho \left(N\right)-\rho \left(N-1\right)$$
$${f\;}^{+}=\rho (N+1)-\rho (N)$$
$$f^{0}=\frac{\left(\rho \left(N+1\right)+\rho \left(N-1\right)\right)}{2}$$

In these equations, ρ(N + 1), ρ(N) and ρ(N − 1) are the electron density in the frontier orbital of the anion, the neutral molecule, and the cation respectively. 

Dual descriptor (DD) is a more accurate descriptor derived from the FF, which reflects the comprehensive effect of *f*^−^ and *f*^+^ [127].
DD=ρ(N+1)−2ρ(N)+ρ(N−1)

The condensed Fukui function (CFF) concentrates electrons in a molecule to each atom, showing the magnitude of the change in electron density on each atom when a molecule loses or gains electrons [128,129]. The higher the CFF *f*^−^ value of an atom, the higher the possibility that an electron will be donated from that atom. 

The FF has been widely used in the prediction of the reactive site and appears to be valuable in this respect [130,131]. Houria Djeradi et al. found a significant correlation between the FF and the antioxidant activity of 36 flavonoids [132].

#### 3.1.9. Atom Charge

Atomic charge represents the point charge situated at the center of an atom. It stands as one of the simplest and most intuitive methods to depict the charge distribution in molecules [133]. Currently, multiple methods have been used to calculate atomic charge. Significant disparities exist in the outcomes obtained with these methods. Therefore, the appropriate method must be selected after considering the underlying principles and unique characteristics of each computational method [134].

In the study on the antioxidant activity of flavonoids, the analysis of atomic charge can be used to predict reaction sites and to describe the electrostatic interactions in molecular docking, molecular dynamics simulations as well as Monte Carlo simulations [135,136,137]. The Mulliken charge is currently prevalent in antioxidant research. However, its theoretical foundation is relatively weak, and the outcome is highly dependent on the basic set used. Based on the good results obtained with ligand-receptor interactions in drug design, AM1-BCC charges (Mulliken charge generated by AM1 semi-empirical method, then corrected by bond charge) and MMFF94 charges (Merck Molecular Force Field 94) appear to be more appropriate [138,139].

#### 3.1.10. Redox Potential

Redox potential is considered to be one of the most accurate chemical descriptors of the ability of a compound to gain or lose electrons. Because the redox potential can be determined experimentally relatively easily and unambiguously, it is a relatively rigorous indicator. 

The redox potential of flavonoids is often used in studying their antioxidant behavior. For example, Miličević et al. have studied changes in the electronic structure of 20 flavonoids due to electrochemical oxidation and explored the relationship between the antioxidant activity and oxidation potential of 6 flavonoids more extensively [140]. An example of its use in the last 5 years is the attempt to design multi-target flavonoids to combat Alzheimer’s [141]. 

#### 3.1.11. Density-of-States (DOS)

The DOS function is a tool to visualize the electronic structure of molecules with energy as a variable. It can be adapted to a specific problem, and therefore DOS has numerous forms. When applied to visualize the electronic structure of a molecule, DOS (E) indicates the number of molecular orbitals per energy unit. The DOS versions best suited for studying the antioxidant activity of flavonoids are total DOS (TDOS) and overlap population DOS (OPDOS).

The TDOS curve delineates the density of orbital distribution across different energy regions. TDOS diagrams are valuable for finding all possible sites where bonds can break. Partial DOS (PDOS), also known as ‘local DOS’ or ‘fractional DOS’, elucidates the contribution of specific fragments of a molecule to TDOS.

OPDOS is instrumental in exploring interactions between fragments. Notably, if an OPDOS value is negative within a particular energy range, it indicates that the orbitals within this range promote fragmentation [142,143]. The OPDOS curves stand out as one of the most effective methods for studying the influence of a specific group on the activity of a compound and for constructing structure–activity relationships (SAR).

#### 3.1.12. Donator–Acceptor Map (DAM)

DAM is a tool that directly visualizes the electron-accepting and donating abilities of compounds. In Figure 6, the four distinct zones of the DAM are shown. The value of R_d_ and R_a_ was calculated by dividing ω^−^ and ω^+^ of the compound by ω_F_^−^ (3.40) and ω_Na_^+^ (3.46), respectively [144]. 

Various modified versions of DAMs more suitable for visualizing the antioxidant activity of flavonoids have been developed [145,146]. An example is the full electron–donor–acceptor map (FEDAM) [147]. In FEDAM, electron acceptance (REA) and electron donation (RIE) indices are plotted on the X and Y axis. The REA and RIE are calculated from the VIP and VEA. Another example is the electron and hydrogen donating ability map (eH-DAMA). The Y axis of the map corresponds to ω of a compound, while the X axis corresponds to the BDE of that compound. Compounds located in the bottom left region of the eH-DAMA are likely to be good radical scavengers, as they tend to easily donate electrons (low ω) and hydrogen atoms (low BDE). 

#### 3.1.13. Electron Localization Function (ELF) and Localization Orbital Locator (LOL)

ELF is a three-dimensional real space function with a value ranging from 0 to 1. The upper limit value of 1 indicates that the electron is completely localized in this range. Electrons are easily delocalized from regions with lower ELF values to other regions.
$$ELF\left(r\right)=\frac{1}{1+{\left(\frac{D\left(r\right)}{D_{0\left(r\right)}}\right)}^{2}}$$

The D(r) is given by the following equation:$$D\left(r\right)=\frac{1}{2}\times \sum \left[i\right]N\left(i\right)\times {\left|▽\psi_{i\left(r\right)}\right|}^{2}-\frac{{\left|▽\rho \left(r\right)\right|}^{2}}{8\times \rho \left(r\right)}$$

In this equation, N(i) is the orbital occupation number, σ is the gradient operator, ψ*_i_* is the wave function, and *ρ*(r) is the total electron density function. The 1/2 × ∑[i]N(i) × |σψ_i_(r)|^2^ represents the kinetic energy density function of the system (also called G(r) or gradient kinetic energy density), |σ*ρ*(r)|^2^/(8 × *ρ*(r)) is the Weizsacker functional [148,149].

The D_0_(r) is an artificial reference standard, so ELF is essentially a relative degree of localization rather than an absolute measure.
$$D_{0\left(r\right)}=\frac{3}{10}\times {\left(3\times {pi}^{2}\right)}^{\frac{2}{3}}\times \rho {\left(r\right)}^{\frac{5}{3}}$$

The characteristics of the atomic shell structure, lone pair electrons, and chemical bonds can be visualized simultaneously in the distribution figure of ELF [150,151]. The ELF function curve can also be used to investigate the changes in the total electron density of different shells. It should be noted that the ELF function has a flaw. When r approaches infinity, D(r) will gradually approach 0, so ELF will be equal to 1 at a position far from the molecule, suggesting complete localization, which is obviously incorrect [152].

LOL and ELF are in essence both reflections of different kinetic energy density functions. Schmider and Becke defined the LOL function, γ(r), in 2000 [153] as follows:$$\gamma \left(r\right)=\frac{t\left(r\right)}{1+t\left(r\right)}$$

The parameter t(r) is defined by the following formula:$$t\left(r\right)=\frac{D_{0\left(r\right)}}{\frac{1}{2}\times \sum \left[i\right]N_{i}\times {\left|▽\psi_{i\left(r\right)}\right|}^{2}}$$

t(r) is the ratio of the kinetic energy density of a uniform electron gas to the current actual kinetic energy density of the system, which has orbital unitary transformation invariance. Therefore, LOL will not become 1 when r approaches infinity. Moreover, LOL offers simpler functional forms and more straightforward graphs compared to those generated by ELF. Consequently, despite the currently predominant use of ELF in studies on the antioxidant properties of flavonoids, LOL appears to be more suitable [154,155].

### 3.2. Thermodynamic Analysis

#### 3.2.1. Related Enthalpies of Different Mechanisms for Direct Scavenging of Free Radicals

The mechanisms that have been mainly considered in the direct scavenging of free radicals by flavonoids are (i) hydrogen atom transfer (HAT), (ii) sequential proton loss electron transfer (SPLET), and (iii) electron transfer followed by proton transfer (ETPT). Their reaction equations and enthalpies are given in Table 2.

In the first attempts to evaluate the antioxidant activity of flavonoids, HAT was considered to be the preferred mechanism. Therefore, BDE values were usually calculated to estimate the free radical scavenging potency, and mostly the 4′-OH on the B ring was found to be the primary reaction site [156]. Later on, it was found that antioxidant activity also depended on the reaction medium, especially the polarity and pH of the solvent. Incorporating this indicated that the SPLET mechanism was the most relevant in vivo [157,158,159,160]. Nowadays, the enthalpies of all these three mechanisms are widely used to estimate the antioxidant potency and pinpoint reaction sites of both neutral flavonoid molecules and their deprotonated forms [161,162].

Initially, the overall reaction was only characterized by the tendency to lose one hydrogen and one electron. Gradually, more complete free radical scavenging mechanisms were explored, involving several electron and proton loss sequences [163]. For example, Amic et al. found that the enthalpy of the second proton-coupled electron transfer is lower than that of the first one, indicating that also this second reaction is involved in the antioxidant activity of flavonoids [164]. Mittal et al. calculated and compared the 12 enthalpies of possible scavenging mechanisms for chalcone and suggested that mechanisms containing several steps should be considered [165]. 

#### 3.2.2. The Reorganization Enthalpy (RE) and Hydrogen Abstraction Energy (HAE)

The RE value describes the energy difference between a specific conformation of a molecule and its optimized conformation [166]. It can serve as a predictor of the conformational changes in flavonoids after the hydrogen extraction process [167]. One of the most representative RE indicators is ortho-hydroquinone formation energy (o-HFE). It represents the enthalpy associated with the formation of an ortho-hydroquinone [168]. This process has been extensively investigated as a critical stage of flavonoids to scavenge free radicals [169]. 

The calculation formulas of HAE and BDE share similarities, but their calculation processes differ. The method proposed by Sroka, Z et al. to calculate the HAE of flavonoids is first to calculate the bond dissociation enthalpies of each site in various conformations that have different orientations of the hydroxyl groups. The minimum value is considered the final HAE for each site of flavonoids [170]. Conversely, to obtain the BDE of flavonoids, the lowest energy conformation is first identified, followed by separate calculations of the bond dissociation enthalpy for each site [58].

Another related descriptor is ΔH*f*, which has been used to calculate the heat needed to convert a flavonoid into the corresponding radical [171].

After reviewing the literature, we conclude that the calculation of enthalpy values does contribute to understanding the antioxidant process. Sometimes a correlation is found between a particular enthalpy descriptor and the antioxidant activity of flavonoids [161], but usually this is not the case. Apparently, it is impossible to evaluate the antioxidant process using a single enthalpy indicator. There are still many factors that cannot be ignored, such as kinetic aspects and the influence of the reaction environment. For instance, Morteza Jabbari et al. investigated the reaction rate between naringenin and 2,2-diphenyl-1-picrylhydrazyl radical (DPPH^•^) in water-ethanol solutions ranging from 50% to 90% ethanol and found that a higher ethanol content results in lower reaction rates [172]. Similarly, Malgorzata Musiali et al. discovered that the reaction rate of ten flavonoids with the DPPH^•^ in alcohols was consistently significantly faster than in acidified alcohols or dioxane [173]. 

Another reason why the antioxidant activity of flavonoids does not correlate with a single particular enthalpy descriptor, is that the antioxidant mechanism of then does not follow a single path. Instead, it more likely can follow various parallel reaction paths, that can be intertwined, spanning multiple steps [5]. Moreover, these paths may differ for different flavonoids. 

### 3.3. Kinetic Studies

Transition state (TS) means the intermediate structure formed in a reaction that has the highest energy. The conventional transition state theory (TST) has been widely employed to view the details of a reaction, especially for radical reactions [174,175]. For the reaction of flavonoids with radicals, the gradual change in the structure of the reactants during the reaction also can be depicted using TST [176,177]. 

Current computational chemistry studies on the kinetics of flavonoids’ antioxidant activity primarily focus on calculating rate constants (K) with TST. The reaction rate constant (K) can be calculated using the ΔG^‡^ (Gibbs free energy of activation) found with TST according to the following equation:$$K\left(T\right)=\sigma \times \kappa \times T\times \frac{k_{B}}{h}\times e^{-\frac{{∆G}^{‡}}{RT}}$$

In the equation, κ, h, and k_B_ denote the Wigner coefficient and the Plank and Boltzmann constants, respectively. The parameter σ reflects the reaction symmetry, which represents reaction path degeneracy [178]. 

In general, for flavonoids, the HAT mechanism predominates in the gas phase, whereas the SPLET mechanism is the primary pathway in polar solvents such as methanol and water [176]. In water, the anion of flavonoids tends to get more attention [162]. Calculation of the electron transfer reaction rates of neutral and anionic forms of a flavonoid can indicate which specific reactions are kinetically favored, and thus identify the favorable reaction pathways [179,180]. This is important for elucidating the mechanisms by which flavonoids scavenge free radicals in biological systems.

However, TST can only calculate the reaction rate of flavonoids with a ‘specific’ free radical, and therefore it is not a descriptor of the overall antioxidant activity. This limitation is inevitable, as the structure of each radical and the solvent environment will affect the reaction rate. In addition, the currently widely used TST is not very successful in accurately calculating the reaction rate constant. This is because the calculation of the reaction rate constant requires more in-depth knowledge of other descriptors, such as the potential energy surface [181]. Given that the rate constant can be determined accurately in experiments, further refining TST and validating theoretical outcomes with experimental data is the logical path to take. 

### 3.4. Interaction Analysis

#### 3.4.1. Quantum Theory of Atoms in Molecules (QTAIM)

The QTAIM is a mathematically rigorous theory concerning atomic region division, bonding analysis, and charge analysis. Its theoretical foundation relies entirely on the properties and geometric characteristics of electron density (as a scalar field), thus minimizing the subjectivity of artificial division and yielding highly precise and quantifiable results. For isolated molecular systems, the most conscientious QTAIM analysis program currently available is Multiwfn, which is both powerful and free [142]. Another powerful open-source program capable of QTAIM analysis is Critic 2 [182]. Its primary focus lies in topological analysis related to critical points, while it also incidentally supports various analysis methods based on real space functions. 

Currently, QTAIM analysis stands out as one of the most powerful tools for analyzing intramolecular hydrogen bonds of flavonoids [58,183,184]. The type of intramolecular hydrogen bond (IHB) could be identified by studying the electron density ρ(r) and Laplacian ∇^2^ρ(r) and Hamiltonian kinetic energy H(r) at the bond critical points (BCP). The hydrogen bond interactions can be divided into the following categories: (1) ∇^2^ρ(r) > 0 and H(r) > 0 indicating a weak hydrogen bond; (2) ∇^2^ρ(r) > 0 and H(r) < 0 indicating a moderate hydrogen bond; and (3) ∇^2^ρ(r) < 0 and H(r) < 0 indicating strong hydrogen bond [185]. Another indicator to describe the strength of hydrogen bonds is the hydrogen bonding energy (EHB) [58]. 

The IHBs are present in the most stable conformation of flavonoids. It reduces the antioxidant activity of hydrogen-bond donors (such as the 5-OH, 3-OH and 3′-OH groups in flavonoids) and enhances the antioxidant activity of hydrogen-bond acceptors (such as the 4′-OH group) [163,186], which is considered crucial for scavenging free radicals by flavonoids [33,78,187]. Moreover, it is closely connected to the ESIPT mechanism. Zhong, Y et al. reported that IHB promotes ESIPT in the S_0_ and S_1_ states of flavonoids [188]. Furthermore, Feixiang et al. found that the site where ESIPT occurs, is partly determined by IHB [189]. 

#### 3.4.2. Reduced Density Gradient (RDG) and Noncovalent Interactions (NCI) Analysis

RDG is a dimensionless parameter utilized to characterize electronic nonuniformity in density functional theory. In 2010, Professor Yang introduced the noncovalent interaction (NCI) analysis method based on RDG, to study the interactions between atoms or molecules [190]. While the electron localization function and QTAIM theory are suitable for analyzing chemical bonds, they possess certain limitations when examining non-covalent interactions like hydrogen bonds, steric hindrance, and π–π stacking. NCI analysis is a potent tool to reveal these non-covalent interactions.NCI analysis can intuitively and comprehensively depict the distribution of weak interactions of flavonoid molecules [27,58]. In NCI analysis, the interaction types are visualized using a color gradient, varying from blue to green indicating a gradual weakening of weak interactions, while red denotes increased resistance. An example is given in Figure 7.

#### 3.4.3. Molecular Docking (MD) and Molecular Dynamics Simulations (MDS)

The principles of MD are the ‘lock-and-key theory’ and the ‘inducible fit theory’, which are models to visualize the reversible binding between a ligand and a receptor [191,192]. The strategies can be divided into: Rigid docking: both the conformation of ligands and the receptors are fixed.Semiflexible docking: only the conformation of the ligands can change freely.Flexible docking: the conformation of both the ligands and the receptors can change freely.

Flexible docking offers the closest simulations of the interaction between ligands and macromolecules [193,194].

MD can rapidly simulate molecular interactions such as hydrogen bonds, electronic interaction, and Van der Waals interaction. MDS can acquire diverse positions, velocities, and trajectories of molecules by solving the potential energy and atomic bonds within the system. It is instrumental in further inferring or evaluating receptor–ligand interactions by simulating the lowest energy conformation and kinetic trajectories of the compounds. 

Despite the availability of multiple matching patterns, experimental data indicate that the current molecular docking scoring functions are still insufficiently accurate [68]. Moreover, MD-MDS solely considers physical changes simulates microscopic molecular behaviors, and neglects to account for the breaking and formation of chemical bonds, as well as the precise dynamics of electrons [195,196]. In addition, MDS is both computationally intensive and time-consuming.

Flavonoid antioxidants regulate oxidative stress through numerous pathways. MD might be used to predict the potency of flavonoids on a specific indirect antioxidant pathway such as the inhibition of radical-generating enzymes, the induction of antioxidant enzymes, epigenetic modification, and also qualitatively elucidate the most dominant mechanism. In the past five years, the most common docking objectives of flavonoids were Keap1 [197], Xanthine oxidase (XO) [198,199], NADPH [9,69], peroxiredoxin 5 (HP5) [200], Human tyrosinase-related protein [68], *Escherichia coli* DNA Gyr [201], α-glucosidase [34], β-lactoglobulin (BLG) [198] and DNA [80]. A novel and promising direction is the use of flavonoids bound to polymers, which may hold great potential for the future [202]. It is important to highlight that, when examining the interaction between flavonoids and macromolecules, the emphasis shifts from reaction kinetics and enthalpy, most prominent in the study of free radical scavenging mechanisms, to equilibriums and Gibbs free energy [203,204,205]. 

Moreover, the integration of semiempirical molecular orbital methods and density functional theory (DFT) is developing. Currently, there have been some promising attempts, such as ONIOM [206].

### 3.5. Bioavailability Analysis

#### 3.5.1. Dissociation Constant (pK_a_)

The pK_a_, defined as the negative logarithm of the equilibrium dissociation constant of an acid into its conjugated base and a proton (K_a_), is a crucial property of flavonoids. It determines the charge of the compound in solutions and thus influences for example solubility, lipophilicity, distribution, protein binding, hydrogen-bound formation, and biological activity. Nevertheless, experimental data regarding the pK_a_ values of flavonoids are scarce. Despite continuous progress, the existing experimental data are frequently incomplete or contradictory [207]. This is predominantly caused by the small difference in the pK_a_ of the various hydroxyl groups in flavonoids. The emergence of computational chemistry is a potential remedy for this issue.

The pK_a_ results provided by computational chemistry are based on the thermodynamic cycle shown in Figure 8, using the following equation [208].
$$pKa=\frac{{{∆G}^{*}}_{soln}}{R*T*ln(10)}=\frac{{\Delta G}_{soln}+\Delta n*1.89}{R*T*ln(10)}$$

In this equation, R represents the ideal gas constant (8.314), n denotes the charge difference before and after deprotonation, and ΔG_soln_ signifies the free energy of the dissociation process. 

Alternatively, there is a more convenient method to obtain the pK_a_ of a compound. In some databases, for example, ACD/Percepta and SPARC, the experimental data of the pK_a_ values of a wide variety of compounds have been collected and used to predict the pK_a_ of related compounds. These software predictions of the pK_a_ appear to be accurate [207,209].

Deprotonation of hydroxyl groups in a flavonoid affects its antioxidant activity [210]. Theoretically, a negative correlation between the free radical scavenging activity of flavonoids and their pK_a_ is expected in polar solvents, as low pK_a_ is more conducive to the SPLET mechanism. Nevertheless, there are some studies showing the opposite. For example, Yi Hu et al. found that for licorice flavonoids, a higher pK_a_ value correlated with a higher solvent-mediated antioxidant activity [64]. Apparently, we have not yet found a clear-cut relationship between pK_a_ and free radical scavenging ability, and this is the subject of ongoing research.

#### 3.5.2. Lipid Water Partition Coefficient (LogP and LogD)

The lipophilicity of compounds is typically estimated by its LogP value, which represents the ratio of the concentration of a compound in water versus that in octanol at equilibrium. The partition coefficient generally refers to the concentration ratio of the non-ionized form of the compound. LogP can be determined experimentally, for instance, using reverse phase chromatography or a shaking flask method. Alternatively, it can be calculated, offering advantages such as not requiring compounds of high purity and saving time and costs. The computer algorithms for predicting LogP of compounds have reached a high level of maturity. Examples are the commercial programs Chem3D and ACD lab, and the open-source programs XLOGP3 [211], ALOGPS [212], MarvinSketch [213] and ChemAxon [214].

LogD refers to the sum of the concentrations of all forms of the compound (ionized plus un-ionized) in water with a specific pH relative to that in octanol, at equilibrium. For compounds that cannot donate or accept protons, LogP equals LogD throughout the whole pH range of the water phase. For acids and bases, the fraction of the compound in water that has a charge depends on the pH. In a water/octanol mixture, mostly the charged molecules are almost completely found in the water phase. Therefore, by combining the log P and the pK_a_ value of an acidic flavonoid with the pH of the water phase, the logD of the flavonoid at that pH can be estimated by logD_pH_ = logP − log(1 + 10^pKa − pH^).

Low values of either LogD or LogP indicate high hydrophilicity and poor lipophilicity, which predicts good water solubility and poor lipid solubility. Most flavonoids enter cells through passive transport. This means that to enter a cell, the flavonoid has to diffuse from the watery environment of the cell into the lipid bilayer of the membrane. Therefore, a low LogP or logD value of a flavonoid indicates that the flavonoid does not easily go into the lipid bilayer. Therefore, the Logp value and—probably more accurately—the logD value serve as crucial parameters for the transport of a flavonoid over membranes [135,215]. A too-high logP value might also be problematic, as these compounds will be sequestrated in fat tissue [216]. According to Lipinski’s Rule of 5, an oral drug should have a LogP value < 5 and ideally between 1.35 and 1.8 for good oral absorption.

As stated in the introduction, usually it is assumed that in cell experiments the concentration of a compound in all compartments of the cell is equal to the initial concentration in the culture medium. Of course, this is not the case. Based on their lipophilicity, notably LogD, flavonoids will compartmentalize in the body over different organs, and in cells over different organelles. This means that LogD should be valued along with quantum chemical calculations to obtain a deeper understanding of the antioxidant activity of flavonoids.

#### 3.5.3. Polar Surface Area (PSA)

The PSA or topological polar surface area (TPSA) is defined as the sum of the surface areas of all electronegative atoms within a molecule (for flavonoids primarily oxygen) along with their attached hydrogen atoms. This means that the calculation principle of PSA is quite straightforward. Currently, the most popular methods to calculate PSA use quantum chemical topology (QCT) analysis, for instance in the QikProp program. It is worth noting that PSA values obtained from different methods may vary, and therefore cannot be directly compared [217].

In addition to LogD, PSA is also used to predict membrane permeability. Molecules with a PSA exceeding 140 square angstroms (Å^2^) typically exhibit poor membrane permeability. For molecules to traverse the blood-brain barrier and thereby interact with receptors in the central nervous system, a PSA of less than 90 Å^2^ is generally required [218].

To exert a biological effect, dietary flavonoids have to cross the intestinal membrane. Over the past five years, PSA has increasingly been used to predict the permeability of flavonoids through cell membranes. The PSA of myricetin [219], kaempferol [219], Q [219] and procyanidin B1(PB1) [68] have been calculated and used as supplementary descriptors to estimate their overall in vivo antioxidant potency.

## 4. Concluding Remarks

Over the past five years, various computational chemistry strategies have significantly advanced our understanding of the antioxidant activity of flavonoids. Moreover, in our drive to progress, each of these strategies is continuously refined to develop even more suitable, accurate and detailed calculation methods.

However, due to the flexibility of the reaction pathway, even the ‘simple’ direct scavenging activity of a flavonoid in a biological system cannot fully be described by only one of the parameters given in this overview. As also stated in the introduction, a minor change in the molecular structure of a flavonoid can have a drastic effect on its antioxidant effect, indicating that each flavonoid has its unique antioxidant profile [220]. Therefore, it is essential to integrate the results of multiple parameters to obtain a comprehensive profile that accurately reflects the antioxidant effects of flavonoids. To address this, we propose a convenient way to visualize the comprehensive antioxidant of flavonoids, displaying various computational descriptors simultaneously in a radar map, as exemplified in Figure 9.

Figure 10 gives an overarching view of the computational landscape of the antioxidant activity of a flavonoid incorporating all discussed descriptors. In this, we follow the paradigm coined by Frazer and Brown that the biological activity of a compound is a function of its molecular structure. Therefore, we have placed the structure activity relationship (SAR) in the center. By integrating the antioxidant profiles of structurally related flavonoids in a SAR, clues can be found to identify the structural elements of the flavonoid molecule that are important for its antioxidant activity, as well as the type of interaction, and the consequence of this interaction, thus enhancing our understanding of the molecular mechanisms underlying flavonoid antioxidant activity, as well as their biological effects and health implications.

## Figures and Tables

**Figure 1 molecules-29-02627-f001:**
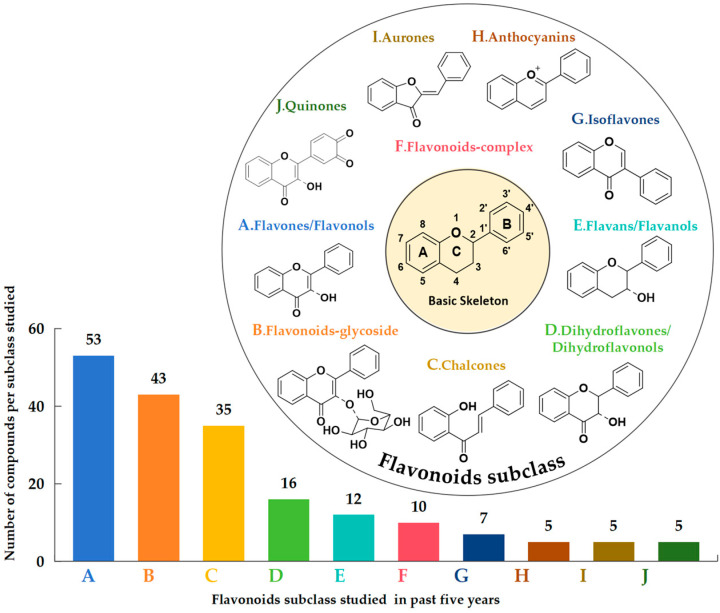
The basic skeleton of the subclasses to which the investigated compounds belong, along with the number of compounds in each subclass that have been studied over the past five years [9,10,11,12,13,14,15,16,17,18,19,20,21,22,23,24,25,26,27,28,29,30,31,32,33,34,35,36,37,38,39,40,41,42,43,44,45,46,47,48,49,50,51,52,53,54,55,56,57,58,59,60,61,62,63,64,65,66,67,68,69,70,71,72,73,74,75,76,77,78,79,80,81,82]. In Appendix A, the compound names and corresponding references are given.

**Figure 2 molecules-29-02627-f002:**
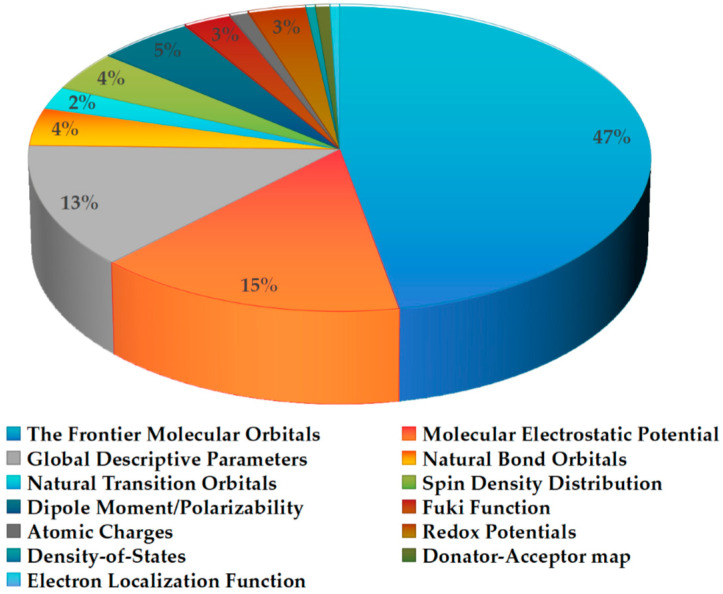
The methods employed in the past five years to investigate the electronic structure of flavonoids, along with the proportion of each method used [9,10,11,12,13,14,15,16,17,18,19,20,21,22,23,24,25,26,27,28,29,30,31,32,33,34,35,36,37,38,39,40,41,42,43,44,45,46,47,48,49,50,51,52,53,54,55,56,57,58,59,60,61,62,63,64,65,66,67,68,69,70,71,72,73,74,75,76,77,78,79,80,81,82]. (The percentages less than 2% are not marked here). In Appendix A, the compound names and corresponding references are given.

**Figure 3 molecules-29-02627-f003:**
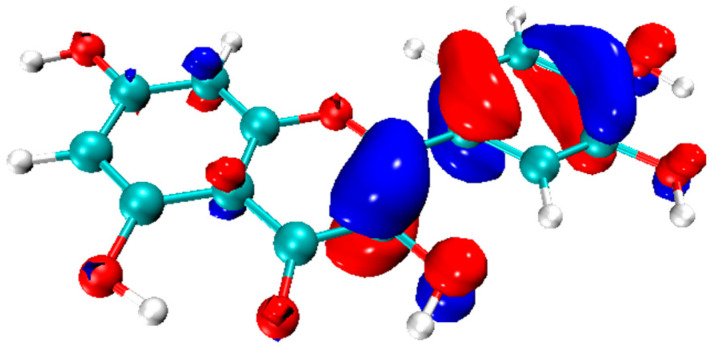
The HOMO map of Q. Calculated with Gaussian 09 package [86] using the DFT method at the M062X [87]/6-311G (d,p) [88] level.

**Figure 4 molecules-29-02627-f004:**
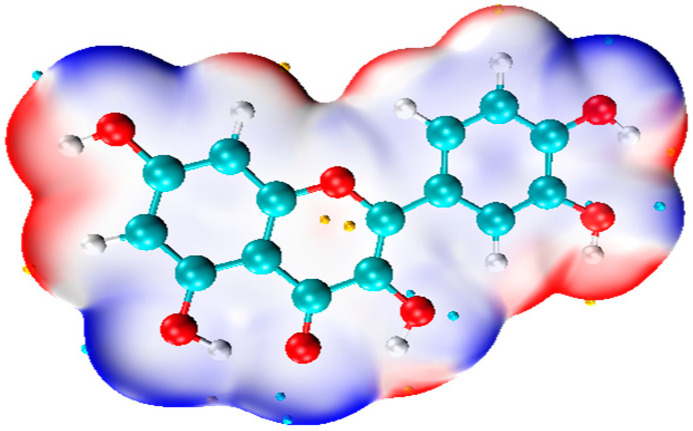
The surface MEP map of Q. Calculated with Gaussian 09 package [86] using the DFT method at the M062X [87]/6-311G (d,p) [88] level.

**Figure 5 molecules-29-02627-f005:**
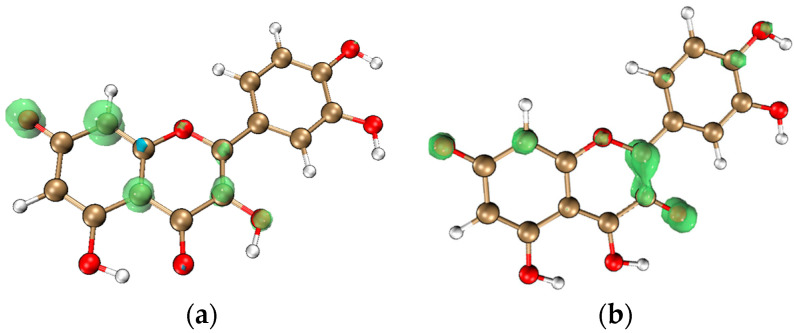
The SDD of the lone pair electron in the Q-7-*O*-radical: (**a**) before and (**b**) after intramolecular proton transfer from the oxygen at the 3-OH to the oxygen at the 4-carbonyl. Calculated with Gaussian 09 package [86] using the DFT method at the M062X [87]/6-311G (d,p) [88] level.

**Figure 6 molecules-29-02627-f006:**
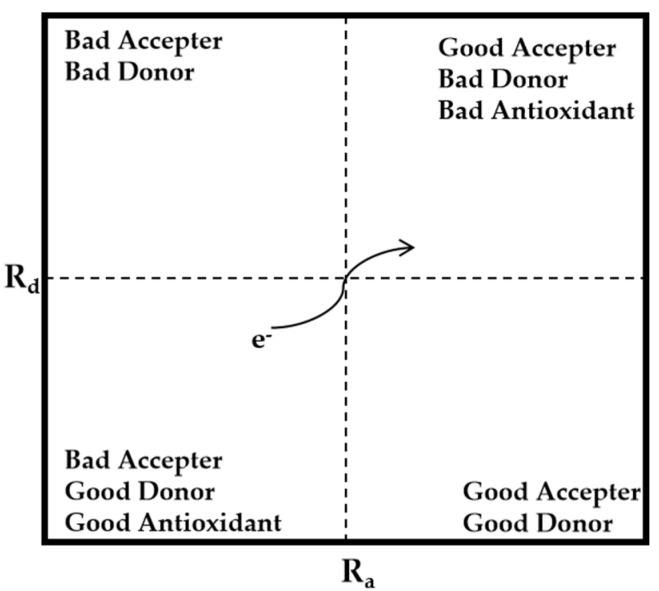
Schematic diagram of the DAM.

**Figure 7 molecules-29-02627-f007:**
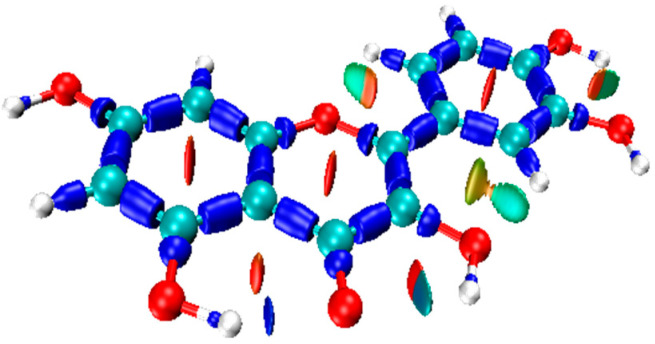
The NCI analysis map of Q. Calculated with Gaussian 09 package [86] using the DFT method at the M062X [87]/6-311G (d,p) [88] level.

**Figure 8 molecules-29-02627-f008:**
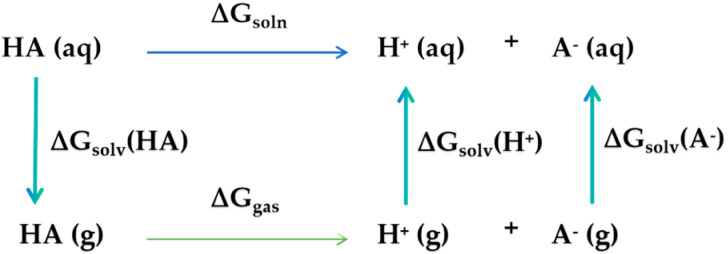
The thermodynamic cycle connects the deprotonation of the acid HA dissolved in water (aq) to that as a gas (g). The values of ΔG_gas_(H^+^) and ΔG_solv_(H^+^) are known from experiments (with a sum of 269.0 Kcal/mol). The other gas-free energies and the solvation-free energies require relatively complex calculations using experimental data and different basis sets.

**Figure 9 molecules-29-02627-f009:**
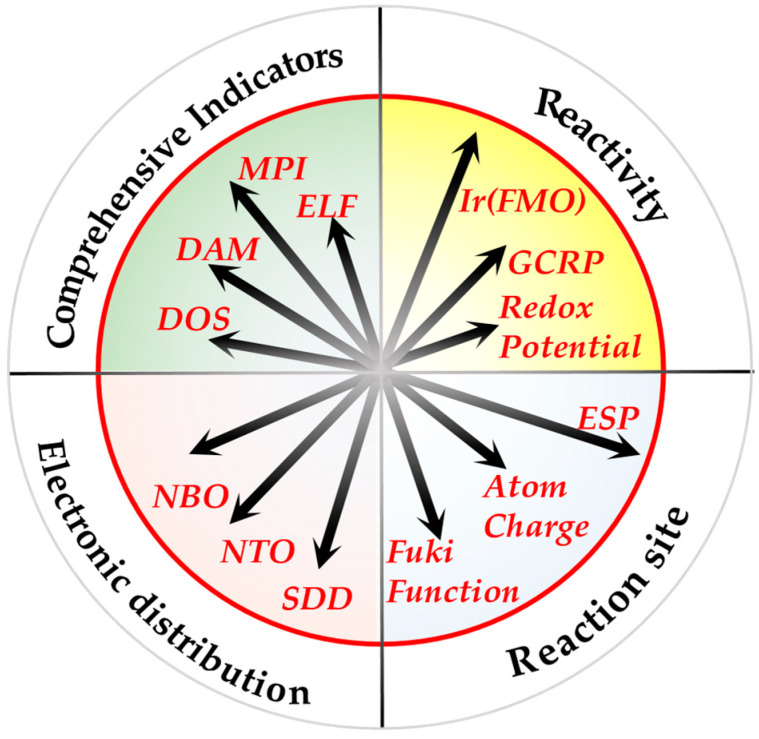
A way to visualize the comprehensive antioxidant profile of a flavonoid. The various electronic characteristics of an antioxidant are related to those of the very potent antioxidant flavonoid Q. The radius of the red circle is the value of each descriptor for Q. The length of each vector starting from the center of the circle can be used to visualize the value of the denoted descriptor of a particular flavonoid relative to that of Q.

**Figure 10 molecules-29-02627-f010:**
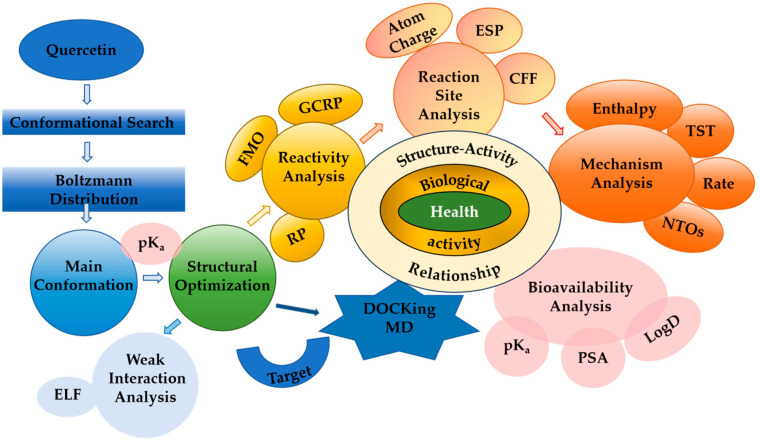
An overarching view on the computational landscape of the antioxidant activity of Q.

**Table 1 molecules-29-02627-t001:** The conventional GCRPs measure the electron transfer characteristics of antioxidants.

Properties	Formula	Description
Hardness(η) Softness (S)	η = (IP − EA)/2 S = 1/2η	η reflects the reluctance towards deformation or polarization of the electron cloud under slight perturbation [106,107,108].
Chemical potential (μ)	−μ = (IP + EA)/2	μ indicates the direction of charge flow. Electrons will migrate from high μ to low μ locations [109,110].
Electronegativity (χ)	χ = (IP + EA)/2	χ gives the tendency of a molecule to attract electrons.
Electrophilicity (ω)	ω = μ^2^/2η	ω reflects the electron-donating ability of a molecule.
ω^+^	ω^+^ = (IP + 3EA)^2^/16(IP − EA)	ω^−^ and ω^+^ reflect respectively the electron-donating and electron-accepting ability of antioxidants [111,112].
ω^−^	ω^−^ = (3IP + EA)^2^/16(IP − EA)	

**Table 2 molecules-29-02627-t002:** The reaction equations and enthalpies of the three main mechanisms of flavonoids to directly scavenge free radicals.

Mechanism	Reaction Equation	Enthalpy
HAT	ArOH_optimized_ → ArO^•^_optimized_ + H^•^	Bond dissociation enthalpy (BDE)
SPLET	ArOH_optimized_ → ArO^−^_optimized_ + H^+^	Proton affinity (PA)
	ArO^−^_unoptimized_ → ArO^•^_optimized_ + e^−^	Electron transfer enthalpy (ETE)
ETPT	ArOH_optimized_ → ArOH^•+^_optimized_ + e^−^	Ionization potential (IP)
	ArOH^•+^_unoptimized_ → ArO^•^_optimized_ + H^+^	Proton dissociation enthalpy (PDE)

## Data Availability

The data presented in this study are available on request from the corresponding author.

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
