# Peer review of "Computational Chemistry Strategies to Investigate the Antioxidant Activity of Flavonoids—An Overview"

_molecules, 2024, doi:10.3390/molecules29112627_

Round 1

Reviewer 1 Report

Comments and Suggestions for Authors

This is a review of computational (mostly quantum-chemical) modelling of the antioxidant behaviour of a few classes of flavonoids. A variety of descriptors that have been used in the literature for this purpose are presented, critically discussed and, in most cases, illustrated with some representative examples. An impressive amount of primary literature is covered, and the paper is well conceived, structured and written. The conclusion that no single calculation (or descriptor) is able to predict the target property well enough is perhaps a bit less optimistic than one would have hoped, but is well supported by the case studies covered. Overall, I think the authors have done a great job putting all this together, and it is a pleasure to recommend the paper for acceptance.

There is one major comment, however, and several minor ones, that should be addressed in a revised version: My major reservation concerns the distinction between BDE and HAE (cf. Fig. 7 on p. 11). The BDE has a well-defined meaning and is an experimental observable, unlike the reorganisation energy and the HAE in Fig. 7, which are artifacts of an incomplete optimization (notwithstanding their interpretative value). In my opinion, the labels BDE and HAE should be swapped in that figure (and in Tab. 3), and HAE better called “frozen” HAE or something, to make clear it involves unrelaxed structures. If the present use is common in the literature covered, a critical comment along these lines is in order.

Minor points:

p. 3, Fig. 2: The quality (resolution) needs to be improved.

p. 4, l. 98: Acronyms DOS and DAM need to be explained. Personally I am not clear about the usefulness of DOSs for molecular systems, this is a property important for bulk materials.

p. 4f., Figs 3 and 4: The computational level should be indicated.

p. 5, l. 147: I don’t understand the concept of a “local ionization energy” – IE is a difference between electronic energy levels, irrespective of the localization of the orbitals involved – elaborate!

p. 5, l. 176: Koopman’s theorem is explained inaccurately. It is an approximation to IP and EA from MO energies inherent to HF theory (a cautionary note is in order that this is actually poorly defined in DFT – no problem of course if adiabatic or vertical IPs are calculated from two separate calculations for neutral and cation).

p. 6, l. 183: The acronym GRCP should be spelled out (it does not conform to the title of Tab. 2)

p. 8, l. 276: The assertion that exactly “12 methods” have been used to calculate atomic charges begs the question, which ones? Maybe just say, “a number of methods”. The acronyms AM1-BCC and MMFF94 need to be explained. ESP-derived charges (e.g. CHELP) used to be popular in the biochemical community, this may be worth a comment.

p. 12, l. 449: The “B” in “kB” in the formula needs to be subscript. It would be best to denote the free energy of activation with G with a superscript double dagger instead of Ga (which looks like an element symbol…)

p. 13, l. 485f: The “2” after the Nabla operators must be superscript (to denote the square).

p. 13, Fig 8b: The quality (resolution) needs to be improved.

p. 14, l. 573: The DeltaG values for the proton from experiment are associated with some uncertainty – an error bar must be specified for the number quoted (269.0 kcal/mol – note the typo un the unit in the text, “Kcal”).

p. 15, l. 576: Where does the factor 1.89 in the equation come from?

p. 19, chapter 3.5.2: It would be good to quote some illustrative logP and log D values.

p. 18, l. 711ff: The sentence about the proton consisting of “small particles that miraculously and spontaneously can covert into other particles depending on the perspective taken” is nonsensical. A proton is just one particle that can’t convert into anything else unless subject to conditions applied in nuclear physics. It can (and will) attach to other species around, notably water, but I don’t see anything miraculous in that (or what perspective this could depend on). Best to just scrap that sentence.

Reviewer 2 Report

Comments and Suggestions for Authors

The authors presented an interesting and thorough review on computational chemistry strategies to investigate the antioxidant activity of flavonoids by analyzing an impressive list of literature. They have tried to do so in a way that is as accessible to the reader as possible. In my opinion the manuscript can be published in the journal. However, a thorough revision is necessary. Please, see my suggestions.

A grammar check is required. The text has sentences that are difficult to read and incomprehensible, as well as typos.

Lines 65-66: “less ambiguous strategies as the one used previously”. It is not clear from the text above what strategies were used. It would be useful to briefly discuss these strategies, their benefits and drawbacks, providing relevant references. In this aspect, all the advantages of computational chemistry will become obvious.

The resolution of the figures is quite low in some cases, so the images are hard to read.

Comments on the Quality of English Language

 Extensive editing of English language required

Round 2

Reviewer 2 Report

Comments and Suggestions for Authors

The authors have carefully revised the manuscript. It can now be published in the journal.

Comments on the Quality of English Language

Minor editing of English language required.